# The use of brain-machine interface, motor imagery, and action observation in the rehabilitation of individuals with Parkinson's disease: A protocol study for a randomized clinical trial

**Kátine Marchezan Estivalet[1], Tatiana Salayaran de Aguiar Pettenuzzo[1], Natália Lopes Mazzilli[1], Luis Fernando Ferreira** [1,2]*, **Fernanda Cechetti[1]**

**1** Federal University of Health Sciences of Porto Alegre, Porto Alegre, Brazil, **2** Queen's University of Belfast, Belfast, Northern Ireland, United Kingdom

* proffernandof@gmail.com

## Abstract

### Background

Parkinson's disease (PD) is a neurodegenerative condition that impacts motor planning and control of the upper limbs (UL) and leads to cognitive impairments. Rehabilitation approaches, including motor imagery (MI) and action observation (AO), along with the use of brain-machine interfaces (BMI), are essential in the PD population to enhance neuroplasticity and mitigate symptoms.

### Objective

To provide a description of a rehabilitation protocol for evaluating the effects of isolated and combined applications of MI and action observation (AO), along with BMI, on upper limb (UL) motor changes and cognitive function in PD.

### Methods

This study provides a detailed protocol for a single-blinded, randomized clinical trial. After selection, participants will be randomly assigned to one of five experimental groups. Each participant will be assessed at three points: pre-intervention, post-intervention, and at a follow-up four weeks after the intervention ends. The intervention consists of 10 sessions, each lasting approximately 60 minutes.

### Expected results

The primary outcome expected is an improvement in the Test d'Évaluation des Membres Supérieurs de Personnes Âgées score, accompanied by a reduction in task execution time. Secondary outcomes include motor symptoms in the upper limbs, assessed via the Unified Parkinson's Disease Rating Scale - Part III and the 9-Hole Peg Test; cognitive

**Data availability statement:** No datasets were generated or analysed during the current study. All relevant data from this study will be made available upon study completion.

**Funding:** The author(s) received no specific funding for this work.

**Competing interests:** The authors have declared that no competing interests exist.

function, assessed with the PD Cognitive Rating Scale; and occupational performance, assessed with the Canadian Occupational Performance Measure.

## Discussion

This study protocol is notable for its intensive daily sessions. Both MI and AO are low-cost, enabling personalized interventions that physiotherapists and occupational therapists can readily replicate in practice. While BMI use does require professionals to acquire an exoskeleton, the protocol ensures the distinctiveness of the interventions and, to our knowledge, is the first to involve individuals with PD.

## Trial registration

ClinicalTrials.gov NCT05696925.

## Introduction

Parkinson's Disease (PD) is a complex neurodegenerative disorder characterized by both motor and non-motor symptoms, progressively incapacitating the individual [1]. PD leads to impairments in motor skills, with multiple manifestations including tremor, rigidity, bradykinesia, and postural instability [2,3]. In the upper limbs (UL), PD negatively affects motor control and planning, resulting in disturbances in dexterity and both gross and fine motor skills, which interfere with the execution of daily activities, such as maintaining self-care, performing productive tasks, and participating in leisure activities [1,4].

Additionally, there is a relationship between the motor symptoms of the UL and cognitive demand, revealing deficits in reach, grasp, and action speed [5]. The greater the motor impairment in the UL, the more significant the impairment in performing cognitive tasks [6]. This raises concerns given the prevalence of dementia and mild cognitive impairment in PD, with issues spanning several domains, including concentration and attention, working memory, recent events, as well as difficulties with calculations, spatial orientation activities, and executive functions [1,7,8]

In light of these disabling symptoms, there has been a focus on incorporating additional approaches to optimize rehabilitation gains which aim to maximize movement quality through personalized interventions linked to the stage of PD progression [9], such as motor imagery (MI) and action observation (AO) [10]. MI involves the mental rehearsal of motor acts, where the individual simulates motor actions and sensations internally [11]. To enhance the neuroplasticity promoted by MI, the brain-machine interface (BMI) emerges as a new technology that decodes neural signals in real-time via electroencephalogram (EEG) during MI practice, activating external devices such as prosthetics or robots and providing instant feedback to the attached limb [12].

AO is a technique based on the activation of the mirror neuron system, referring to a group of neurons distributed throughout the cerebral cortex that activate similarly when observing an action and when executing it. Thus, by observing an action, the brain can map the movement, creating a motor representation and executing it internally, thereby acquiring knowledge and motor memory [13,14]. Both AO and action execution (AE) can promote motor learning, facilitating cortical reorganization and restoring cognitive references. However, there is no detailed description of the protocols applied for these techniques [15].

Therefore, due to the motor impairments caused by PD in the UL and the accompanying cognitive function deficits, along with the scarcity of studies involving MI, AO, and BMI

- both individually and as combined techniques in PD - and the lack of detailed descriptions of the applied protocols, there is a clear need for additional research to develop innovative rehabilitative approaches and further reinforce the existing evidence with larger studies. Hence, this protocol study aims to provide a description of the rehabilitation protocol that can be utilized to evaluate, through a simple blinded randomized controlled clinical trial, the effects of the isolated and combined application of MI and AO, along with BMI, on motor alterations in the UL and cognitive function in PD.

## Methods

### Study design

This is a description of a protocol for a simple blinded randomized clinical trial to evaluate the effects of the application of MI, AO, and BMI, either individually or in combination, on motor alterations in the UL of individuals with PD. All participants will be evaluated pre-intervention (T0) and post-intervention (T1), and will undergo a follow-up (T2) four weeks after the end of the intervention.

The clinical trial was designed following the Consolidated Standards of Reporting Trials (CONSORT) [16]. The protocol follows the item definitions for clinical trials as per the Standard Protocol Items: Recommendations for Interventional Trials - SPIRIT [17] (S1 Checklist).

### Study setting

The data will be collected on the premises of the Federal University of Health Sciences of Porto Alegre, in Porto Alegre, Brazil.

### Ethical aspects

The study is registered on clinicaltrials.gov with the study ID number: NCT05696925 (S3 File). The project was approved by the Research Ethics Committee of the Federal University of Health Sciences of Porto Alegre (UFCSPA), under letter nº 5.638.729, and received the Certificate of Ethical Appreciation Presentation (CAAE) number 61710822.0.0000.5345 (S1 File and S2 File).

To be included, volunteers must read and sign the Informed Consent Form. The entire research was written following Resolution 466/2012 of the Brazilian National Board of Health, and adhered to the principles of the Declaration of Helsinki for research involving human subjects. Data will be processed in accordance with the General Data Protection Law (Brazilian Law nº 13.709/2018).

All researchers are aware of and agree to maintain the confidentiality of personal information about participants that will be collected, shared, and maintained before, during, and after the trial, having signed the Confidentiality Agreement submitted along with the UFCSPA Ethics Committee, and access to the final dataset of the trial will be limited solely to the researchers.

### Eligibility criteria

**Inclusion criteria.** Have a diagnosis of PD and be at stages 1–3 on the Hoehn and Yahr scale [18]; be at least 20 years old; be on stable medication; achieve a score higher than 23 on the Montreal Cognitive Assessment (MoCA) [19]; score a minimum of 20 points on the Kinesthetic and Visual Imagery Questionnaire (KVIQ-10) [20,21]; exhibit motor impairment in the dominant UL, with scores above average according to manual dominance and sex in the 9-Hole Peg Test (9HPT) [22]; and have signed the Informed Consent Form.

**Exclusion criteria.** Having additional central nervous system disorders or other conditions that could affect upper and lower limb function; having other uncontrolled chronic conditions that may interfere with participant safety; and not be using deep brain stimulation (DBS) devices.

## Participant timeline

The schematic diagram (Fig 1) outlines the overall timeline and engagement duration for participants in the trial, detailing the time allocated for each phase, starting from the initial eligibility screening to the conclusion of the study. It includes the time periods during which the trial interventions will be administered, as well as the procedures and assessments conducted in each phase.

## Randomization, allocation and blinding

Following selection, participants will be randomly allocated into one of five experimental groups: GE1 (motor imagery, action observation, and action execution), GE2 (motor imagery and action execution), GE3 (action observation and action execution), GE4 (motor imagery via brain-machine interface and action execution), and GE5 (motor imagery via brain-machine interface, action observation, and action execution) (Fig 2).

Allocation will be carried out by a blinded investigator who will not be involved in the intervention. The randomization process will be carried out with a 1:1 allocation ratio through the website randomizer.com. It will not be possible to blind the participants and researchers who will carry out the training. However, the study will be blinded to the researchers for results and statistical analysis. This is a single-blind study, as the therapists (physical and occupational therapists) conducting the interventions will be different from those responsible for

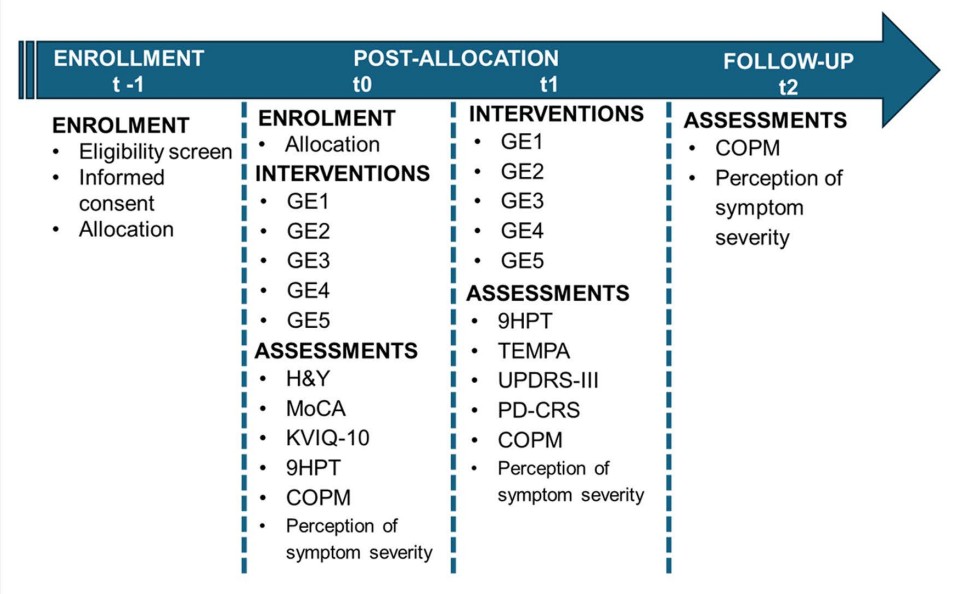

**Fig 1. SPIRIT schedule of enrolment, interventions, and assessments of the study.** COPM, Canadian Occupational Performance Measure; H&Y, Escala Hoehn **e** Yahr; KVIQ-10, Kinesthetic and Visual Imagery Questionnaire; MoCA, Montreal Cognitive Assessment; PD-CRS, Parkinson's disease-cognitive rating scale; t, time; TEMPA, Test D'évaluation Des Membres Supérieurs Des Personnes Âgées; UPDRS, Unified Parkinson's Disease Rating Scale; 9HPT, Nine Hole Peg Test.

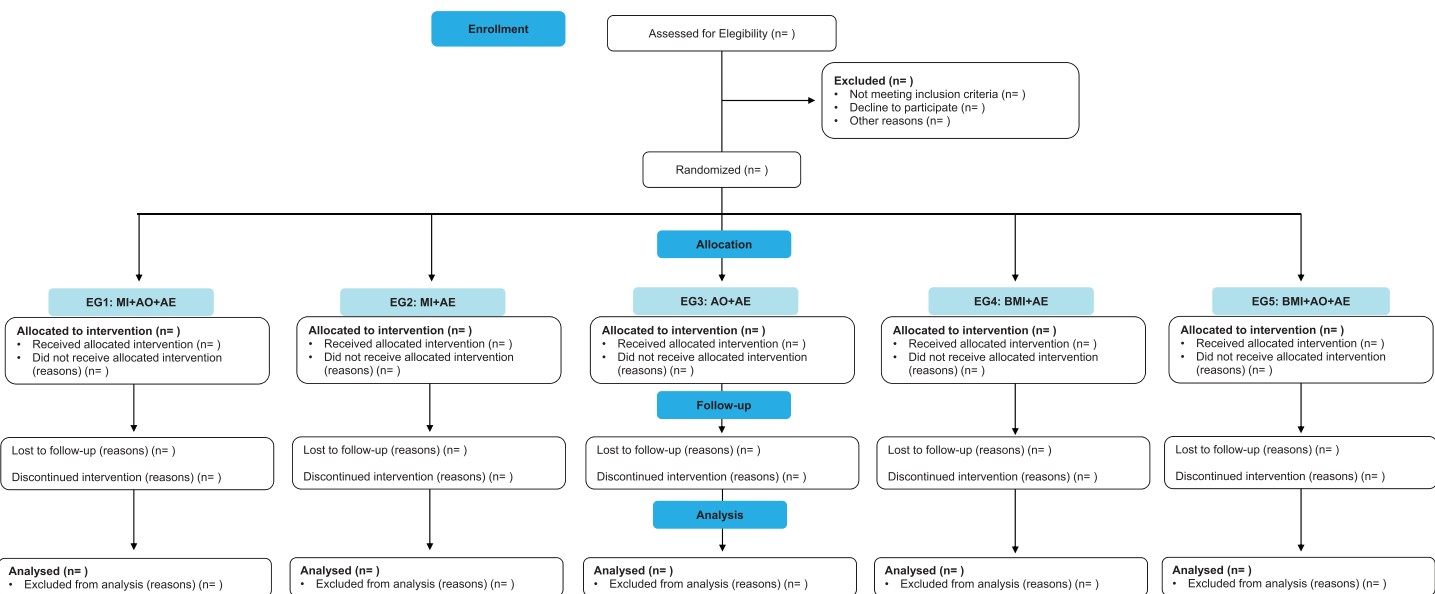

**Fig 2. Flowchart of participant distribution throughout each stage of the study.** AO, action observation; BMI, brain machine interface; AE, action execution; EG, experimental group; MI, motor imagery.

administering the assessment tools during data collection phases. Additionally, the blinding of data analysis will be ensured.

## Data collection

The following instruments will be used for pre-intervention (T0) and post-intervention (T1) assessments. The Test d'Évaluation des Membres Supérieurs de Personnes Âgées (TEMPA) will be employed to evaluate limitations of the UL through tasks that simulate activities of daily living [23]. The final score of the test, which can reach up to 186 points, corresponds to the sum of the functional score, encompassing unilateral and bilateral tasks (totaling up to 36 points), and the task analysis score, which assesses five areas of sensory and motor skills in the UL (totaling up to 150 points). In this study, unilateral tasks will be performed using the more affected UL. For better interpretation, especially in task analysis, sessions will be recorded for subsequent analysis.

To assess fine manual motor performance, the 9-Hole Peg Test (9HPT) will be administered, considering the average times based on sex and manual dominance: for men, 21.1 seconds for the dominant hand and 22.3 seconds for the non-dominant hand; for women, 19.9 seconds for the dominant hand and 21.4 seconds for the non-dominant hand [22]. This instrument will be used as a screening tool, and data from included participants will be analyzed in the final results.

To evaluate motor disturbances in PD, the Unified Parkinson's Disease Rating Scale - part III (UPDRS-III) will be utilized, which involves motor assessment [24], excluding items related to facial symptoms or those corresponding to motor symptoms in the lower limbs. The maximum score is 32 points, with higher scores indicating worse symptoms.

Cognitive functions will be assessed using the Parkinson´s disease -Cognitive Rating Scale (PD-CRS), which encompasses nine functions: sustained attention, working memory, immediate and delayed verbal memory recall, alternating verbal fluency, action verbal fluency, spontaneous clock drawing, visual confrontation naming, and clock drawing copying. The total score for the PD-CRS is 134 points [25].

The Canadian Occupational Performance Measure (COPM) is a semi-structured interview that identifies problem activities in the areas of self-care, productivity, and leisure. For this study, participants will choose up to five problem activities, rating them from 1 to 10 based on their performance and satisfaction [26]. In cases of motor symptoms in the non-dominant UL, participants will be instructed to select problem activities that involve more bimanual actions.

In addition to the previously mentioned instruments, the researchers will ask participants about their perception of severity with the following questions: 1) "How would you rate the severity of your upper limb motor functions (dexterity, speed, movement precision)?"; 2) "How would you rate the severity of your cognitive functions (memory, attention, concentration, organization)?". Participants will respond using a Likert scale: none, mild, moderate, severe, and extreme.

For the follow-up assessment (T2), conducted four weeks after the completion of the interventions, the COPM will be reapplied, along with the questionnaire regarding the perception of severity of motor and cognitive functions.

## Interventions

Each participant will undergo the first stage of data collection, the pre-test (T0), which will take place one day before the treatment period begins. This stage will consist of the administration of the four assessment instruments. Additionally, on this same day, participants assigned to the GE4 and GE5 groups will experience the interventions to familiarize themselves with how they will be conducted throughout the treatment.

The next stage involves the actual intervention. Regardless of the allocated group, all participants will undergo 10 treatment sessions, with each session lasting approximately 60 minutes. The entire study process will span a period of two weeks, with a two-day interval in the middle [27].

It is important to note that the activities to be imagined, observed, and executed during the interventions will be pre-selected by the therapist based on the participant's preferences indicated through the COPM. The choice of activity will also depend on the dominance of the affected limb; if the dominant side is more affected, the activity will preferably be unilateral. However, if the more affected side is the participant's non-dominant side, the activity will necessarily be bilateral. To ensure equal application of the interventions across all experimental groups, the techniques will be administered by the same therapist according to the following protocol:

**Motor imagery (MI).** The activities will be narrated by the therapist during the intervention, guiding the participant - eyes closed and seated - to imagine executing the action from a first-person perspective, without the presence of motor symptoms and without performing any actual movement. To facilitate imagination, the participant may hold or grasp a real object related to the activity being imagined, either before or during the imagery, serving as a sensory stimulus for motivation. At the end of each imagined activity, the therapist will ask the participant whether they were able to imagine themselves performing the requested activity without motor symptoms. Each imagery session will last approximately 2 minutes.

**MI via brain-machine interface (BMI).** The Neurobots equipment will be utilized (Exobots System Software: 1.10.0, Exobots Firmware version 2, EEG Firmware version 1, Neurobots, Pernambuco, Brazil). On the first day, the participant will be registered in the software system, and the therapist will provide a detailed explanation of the intervention, followed by a 2-minute training session with the Exobots system to ensure adequate understanding.

During the intervention, the participant will sit and wear a cap adjusted to their head circumference, containing EEG signal acquisition electrodes in the regions FC3, C3, CP3, FC4, C4, and CP4 (Fig 3). The electrode positioning will follow the international 10–10 EEG mapping standard. To ensure proper neural signal capture and low transmission impedance, the electrodes will be filled with conductive gel [28].

An exoskeleton will be attached to the more affected UL using Velcro straps on the forearm (1), wrist (1), and fingers (5). The participant will be instructed to keep their eyes closed, with their arm in flexion and abduction, elbow flexed at 90°, forearm pronated, wrist in a neutral position, and the hand with the exoskeleton resting on a cushion.

Each of the 10 intervention sessions will consist of 10 exercises with 5 repetitions (including the blocks "relax," "prepare," "think," and "move"), with the 5 chosen tasks imagined twice. Each exercise will last 2 minutes, with a 30-second break between them for the participant to rest. The approximate duration of this stage will be 30 minutes. During each repetition, the participant will listen to four audio commands from the software system: "relax," "prepare," "think," and "move." When hearing "Relax", the participant should think of nothing; during "Prepare" and "Think", the participant will imagine the action, and during "Move", the exoskeleton will automatically open and close the participant's hand.

In the "relax" block, the software will quantitatively display this state on the computer screen through a feedback bar, indicating the absence of thought (a clear mind). In the "prepare" and "think" blocks, if the participant can maintain thought for at least 3 seconds (the established time window to avoid false positives) and achieve a score of 70 points (the minimum threshold shown on the feedback bar), the exoskeleton will move (opening and closing the participant's hand), indicating that the system has detected continuous activation in the primary motor cortex, premotor cortex, or primary somatosensory cortex.

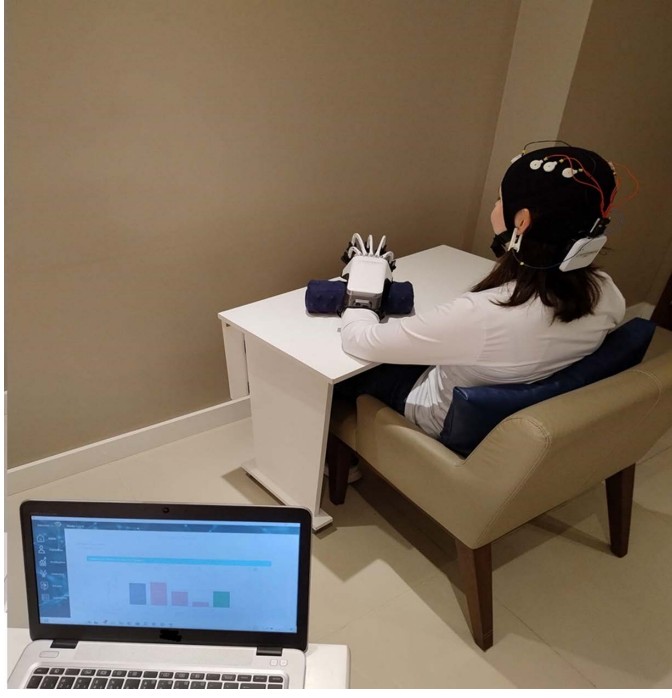

**Fig 3. BMI equipment system.**

At the end of each exercise, performance will be verified through a graph provided by the system, displaying the score of the achieved degree of imagined movement alongside the most activated brain areas mentioned earlier. Upon completing the 10 intervention sessions, the system will display a graph showing the average degree of imagined movement for each session.

At the beginning of each session, it will be checked whether all 6 electrodes are active and if they detect any noise (wet air, metal objects, electronic devices, strong muscle contractions) that could interfere with the capture of the alpha wave signals in the brain. The manufacturer has established a 70% reduction (i.e., 70 points) in low-frequency activation (8–32 Hz), known as event-related desynchronization (ERD), as the activation threshold, an intermediate value between the observed percentage ranges [29].

**Action observation (AO).** Participants will watch previously recorded videos by the therapist, in healthy conditions, from first and/or third-person perspectives, featuring front or side views that demonstrate the execution of activities. The AO will be conducted while seated, and the recorded videos will not be narrated and will be displayed on a computer screen. During the AO, participants will be asked to focus their attention on the details of the movements of the UL without making any movements themselves. Each observation session will last approximately 2 minutes.

**Action execution (AE).** Participants will be instructed to perform the activities as imagined and observed, which can be done while seated or standing. They will be encouraged to execute the activities as they are able, with some activities prompting them to perform as quickly as possible or to alternate hand dominance based on motor symptom manifestation, especially in unilateral activities. Each execution session will also last approximately 2 minutes.

Below are presented the applicability sequences of each intervention group according to their respective approaches (Fig 4).

| EG1: MI+AO+AE | EG2: IM+AE | EG3: AO+AE | EG4: BMI+AE | EG5: BMI+AO+AE |
|---|---|---|---|---|
| • Each activity will be imagined, observed, and executed twice. <br> • Sequence: imagine, observe, and execute the activity, and then imagine, observe, and execute the same activity again. | • Each activity will be imagined and executed twice. <br> • Sequence: imagine and execute the activity, and then imagine and execute the same activity again. | • Each activity will be observed and executed twice. <br> • Sequence: observe and execute the activity, and then observe and execute the same activity again. | • Two training cycles will be carried out with the exoskeleton. <br> • The participant will remove the exoskeleton and perform each of the five previously imagined activities twice. | • Two training cycles will be carried out with the exoskeleton. <br> • The participant will remove the device and watch the five videos demonstrating the previously chosen activities twice. <br> • The participant will perform each of the activities twice, as previously imagined and observed. |

For each intervention session, 5 activities will be selected in advance.

**Fig 4. Sequence of applicability of interventions.** AO, action observation; BMI, brain machine interface; AE, action execution; EG, experimental group; MI, motor imagery.

During the intervention sessions, the difficulty of tasks may be adjusted as the participant progresses in execution, by altering the objects used in the activities, such as sizes, weights, resistances, and different shapes, to change the types of grips and pinches, as well as the range of motion, in addition to being prompted to perform as quickly as possible to stimulate movement speed. It is crucial that the environment is free from distractions to promote concentration during the sessions. Additionally, to ensure proper EEG recording, participants will be advised not to attend sessions with wet hair.

After the intervention period, each participant will undergo the second data collection phase, the post-test (T1), which will take place the day after completing the 10 intervention sessions. This phase will involve the administration of the same four evaluation instruments applied at T0, along with the 9HPT. On this day, participants will also be reminded of the third data collection phase, the follow-up (T2), scheduled to occur four weeks after the conclusion of the interventions. In the follow-up, participants will receive a call and will again respond to the COPM, reporting their satisfaction and performance regarding the problem activities, as well as completing a questionnaire on their perception of the severity of motor and cognitive functions (S1 Graphycal abstract).

Below are some sequences of images as examples of activities recorded on video for the AO. Three activities will be described, each involving an area of occupational performance: self-care, productivity, and leisure, detailed with technical descriptions of the executed movements and a narrative manner involving verbal commands for MI. Here are the examples:

The activity of organizing weekly medication (Fig 5) is a personal care activity, categorized under self-care according to the COPM, and involves various actions. We can choose only the action of holding the medication bottle with one hand and opening the cap with the other, unscrewing it. Another option is to select different medications, such as varying sizes and colors of tablets, to organize them into different days of the week, also stimulating cognitive functions. Therefore, we note that even though it is a bilateral activity, if the participant exhibits motor symptoms on the dominant side, the therapist will encourage the use of the dominant side to perform the actions of the activity, such as opening and closing the compartment lids, opening and closing the bottle cap, and holding the tablets. If the participant exhibits motor symptoms on the non-dominant side, the therapist will encourage the use of the contralateral side for the actions.

The activity of hanging clothes on a clothesline (Fig 6) is a domestic task, categorized under productivity according to the COPM, and involves various actions, both unilateral (using clothespins) and bilateral (grabbing and hanging clothes). We can select clothespins of different sizes, materials, and resistances, which require more dexterity in the pinch grip, as well as position the clothesline at varying heights to alter the range of motion in the shoulder and elbow along with different types of clothing.

The activity of playing cards with a deck (Fig 7) is a calm recreational activity, falling under the area of leisure according to the COPM. In the AO intervention session, even while using the same video, we can introduce some variations in how the activity is performed, depending on the individual's condition and the expected difficulty of the task. For example, we can ask the participant to distribute the cards as quickly as possible, encouraging speed of movement.

In the AO intervention session, we can use the entire video (see supplementary material – S1, S2, and S3 Videos) or select specific stages, based on the person's motor condition and the expected difficulty of completing the activity. The videos are recorded from a frontal view, in the third person perspective, with the participant's dominant side being the right side. It can be observed in the activities previously described in the verbal commands that at some points the request for action is to be performed with the dominant hand and at other times with the non-dominant hand. However, during the intervention, the therapist may modify which hand

| Image | Step and Technical description of the movements performed during the execution of the activity in AO. | Verbal command guided to the participant when performing the task during the MI. |
|---|---|---|
| 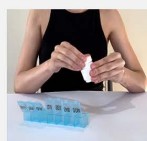 | **a) Opening the medication organizer compartment**<br><br>**DUL:** involves shoulder flexion and abduction, elbow flexion, forearm pronation, wrist extension and flexion movements, and pinching with the thumb and second finger or the thumb, second, and third fingers.<br><br>**NDUL:** holds the medication organizer against the table, keeping the shoulder in flexion and abduction, the elbow supported on the table in flexion, the forearm in pronation, the wrist extended, and the fingers flexed and adducted. | - Close your eyes.<br>- Imagine that you need to organize the medications for the week.<br>- You have a pill organizer with all the days of the week separated and a bottle with the pills of your medication.<br>- With your ND hand, hold the pill organizer, and with your D hand, open the compartments corresponding to each day of the week usin g the tips of your index finger and thumb, starting with Monday, then Tuesday, and continue until you have completed the remaining days of the week.<br>- Notice the texture of the material of the bottle and the pill organizer in your hands. |
| 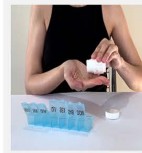 | **b) Open medication bottle**<br>The NDUL holds the bottle with shoulder and elbow flexion, the wrist in a neutral position, and the hand grasped with fingers in flexion and adduction, and the thumb in flexion and abduction. The DUP opens the bottle cap, maintaining shoulder flexion and adduction, elbow flexion, forearm pronation, and a neutral wrist position, performing radial and ulnar deviation movements to unscrew the cap while maintaining hand grasp with fingers in adduction and flexion. | - Now, you need to place a pi ll from the bottle into each day of the week.<br>- With your ND hand, hold the bottle with the medication, and with your D hand, open the cap by twisting it. |
| 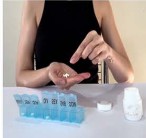 | **c) Removing pills from the jar**<br>The NDUL holds the jar, keeping the shoulder in flexion and abduction, the elbow flexed, and the forearm in pronation. The wrist moves from a neutral position to flexion, and the hand grasps the pills with fingers flexed and adducted, and the thumb flexed and abducted. The DUL maintains shoulder and elbow flexion, supinat ion of the forearm, and wrist extension, with fingers and thumb flexed and adducted to hold the pills in the palm of the hand. | - With your ND hand, hold the medication jar and place some pills in the palm of your D hand, then release the jar onto the table |
| 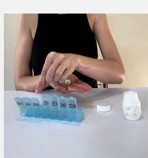 | **d) Picking up pills from the palm of the hand**<br>NDUL – slight abduction and flexion of the shoulder, flexion of the elbow, pronation of the forearm, the wrist moves through flexion and extension, and the hand pinches the pill between the thumb and index finger, keeping the other fingers flexed and adducted. DUL – maintains the same position as in image C. | - Using the ND hand, with the thumb and index finger making a pinching motion, pick up one pill at a time and place it into each compartment for the da ys of the week until all sections of the organizer are filled. |
| 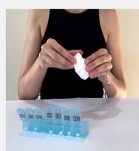 | **e) Placing the pills in the compartments (days of the week)**<br>The NDUL picks up one pill at a time from the palm of the D hand. It maintains shoulder flexion and adduction, performs elbow flexion and extension, pronates the forearm, holds the wrist in a neutral position, and makes a pinching motion using the thumb and index finger. The pinching movements (opening and closing) are repeated until all pills are distributed into the compartments for each day of the week. The DUP maintains the position from images C and D. | - If there are any pills left in your hand, place them back in the bottle, and use the thumb and index finger of your ND hand to pick up the pill from your D hand. |
| 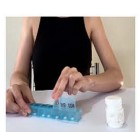 | **f) Closing the medication bottle**<br>The DUL maintains shoulder flexion and abduction, elbow flexion, neutral forearm position, wrist extension, and grasps the bottle. The NDUL maintains shoulder flexion and abduction, elbow flexion, forearm pronation, and performs wrist flexion along with radial and ulnar deviation to screw the cap back onto the bottle, maintaining grip with the hand. | - Then, you take the bottle with your ND hand and pick up the cap with your D hand to close it, then place the bottle back on the table. |
| | **g) Closing the compartments of the medication organizer**<br>The NDUL holds the organizer with shoulder flexion and adduction, a flexed elbow, a pronated forearm, a neutral wrist, and a pinching grip using the thumb with the second and third fingers, while the other fingers remain flexed. The D hand closes each compartment, maintaining shoulder abduction and flexion, a flexed elbow, a pronated forearm, a slightly extended wrist, and flexion-extension movements of the index and third fingers. | - With your D hand, hold the medication organizer, and with your ND hand, close each compartment corresponding to each day of the week, one at a time.<br>- Complete the task. |

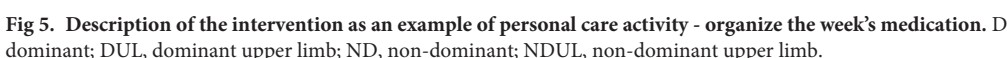

**Fig 5. Description of the intervention as an example of personal care activity - organize the week's medication.** D, dominant; DUL, dominant upper limb; ND, non-dominant; NDUL, non-dominant upper limb.

| Image | Step and Technical description of the movements performed during the execution of the activity in AO. | Verbal command guided to the participant when performing the task during the MI. |
|---|---|---|
| | **a) Holding the clothes in hands** <br> The DUL and NDUL hold the piece of clothing in front of the clothesline. Shoulders are in a neutral position along the trunk, elbows extended, wrists in a neutral position, and hands grasp the clothing with flexion and adduction of the fingers. | - Close your eyes. <br> - Imagine that you need to hang the clothes on the line to dry. |
| | **b) Placing the clothes on the line** <br> Both the D and ND arms lift the clothing to the height of the line, with shoulder flexion and abduction, elbow flexion, forearm pronation, wrist flexion, and the hand grasping the item with finger flexion and adduction. | - Hold the clothing using both hands, but to grab the clothespin, use your D hand. <br> - Pay attention to the movement your shoulder and elbow are making during the task with your ND hand to hold the clothing and with your D hand to pick up the clothespins and bring them to the line |
| | **c) Grabbing the clothespin** <br> The NDUL holds the clothing on the line in the same position as in Figure B. The DUL removes the clothespin from the line, performing shoulder and elbow flexion, forearm supination, wrist extension, and the hand forms a pinch using the thumb and index finger or the thumb with the index and middle fingers to squeeze and open the clothespin to take it off the line. | - Try to notice the movement you make with your fingertips to grab the clothespin and the force you use to squeeze and open it. |
| | **d) Attaching the clothing to the clothesline** <br> The NDUL holds the clothing on the clothesline in the same position as in the previous figures. The DUL attaches the clothing to the clothesline by performing shoulder flexion and horizontal abduction-adduction movements, elbow flexion, forearm pronation, wrist extension -flexion movements, and the hand makes a pinching motion with the thumb and second finger or with the thumb, second, and third fingers to squeeze and open the clothespin, releasing it to secure the clothing to the clothesline. | - Try to notice the texture and shape of the clothespin you are holding. <br> - Also, try to feel the texture of the fabric of the clothing being secured to the clothesline. |
| | **e) Grab another clothespin** <br> The task is repeated with both the DUL and NDUL as shown in figure C, with possible variations, angles, and ranges of motion to remove the clothespin from the clothesline. | - When grabbing the other clothespin, pay attention to the movement you make with your fingers, as well as the rest of your limb (arm). |
| | **f) Clipping the clothes on the line** <br> The NDUL holds the clothes on the line in the same position as in the previous figures. The DUL repeats the movements as in figure D. | After placing and clipping the clothes on the line, finish the task. |

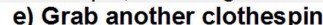
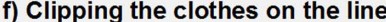
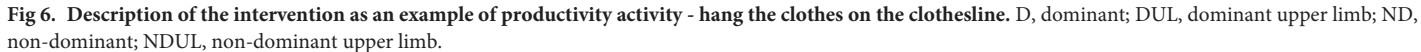

**Fig 6. Description of the intervention as an example of productivity activity - hang the clothes on the clothesline.** D, dominant; DUL, dominant upper limb; ND, non-dominant; NDUL, non-dominant upper limb.

will perform the action, and instead of referring to the dominant and non-dominant hands, they may use the terms right hand and left hand according to the participant's dominance.

All participants will be advised to maintain their activities but not to initiate any treatment or engage in any form of exercise or sport during their participation in the study. Participants are advised to be in the "on" state of their medication for evaluation and intervention. The

| Image | Step and Technical description of the movements performed during the execution of the activity in AO. | Verbal command guided to the participant when performing the task during the MI. |
|---|---|---|
| 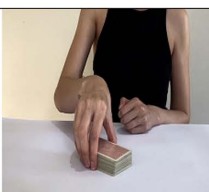 | **a) Picking up cards**<br>The ND hand rests on the table in a neutral shoulder position, with the elbow flexed, forearm in neutral, wrist in neutral, and fingers flexed and adducted. The D arm reaches toward the table to grab the deck of cards, with the shoulder in a neutral position for flexion and abduction. The elbow is flexed, the forearm is pronated, the wrist is in ulnar deviation and flexion, fingers are flexed and adducted, and the thumb is extended and abducted. | - Close your eyes.<br>- Imagine that you are playing cards.<br>- You reach forward to grab the deck of cards with your D hand, which is resting on the table, away from you. |
| 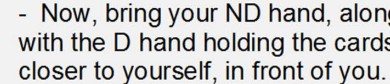 | **b) Grabbing the deck and bringing both hands to the midline**<br>In the NDUL, the shoulder is in a neutral position, the elbow is flexed, the forearm is neutral, and the wrist is in a neutral position, holding the deck of cards with the fingers flexed and adducted and the thumb abducted and flexed. In the DUL, the shoulder is slightly flexed, the elbow is flexed, the forearm is in supination, the wrist is extended with ulnar deviation, and the deck of cards is supported in the palm of the hand with the fingers and thumb flexed and adducted. | - Now, bring your ND hand, along with the D hand holding the cards, closer to yourself, in front of you. |
| 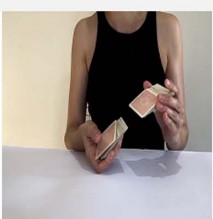 | **c) Shuffling the Cards**<br>NDUL - with the shoulder in a neutral position, elbow flexed, forearm in a neutral position, and wrist extended, shuffle the cards by performing radial and ulnar deviations, with fingers in abduction making small flexion and extension movements as the cards slide into the D hand. NDL, shoulder in a neutral position, elbow flexed, forearm in supination, wrist slightly extended, holding the cards with fingers flexed and the thumb in slight flexion and abduction. | - Now imagine that you begin shuffling the cards, holding part of the cards in your D hand and part in your ND hand, passing them from the D hand to the ND hand.<br>- Notice the texture of the card material in contact with the palm of your hand and your fingers. |
| 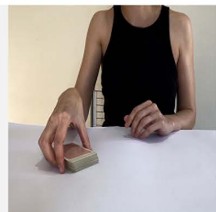 | **d) Placing the cards on the table**<br>NDUL – hand resting on the table in a neutral shoulder position, elbow flexed, forearm pronated, wrist extended, and fingers flexed. DL - shoulder flexed, elbow extended, forearm pronated, wrist flexed and deviated ulnarly, fingers abducted and flexed, and thumb extended and abducted to pick up the cards. | - After shuffling the cards, you place the deck on the table with your D hand, and with the same hand, you split the deck in half. |
| 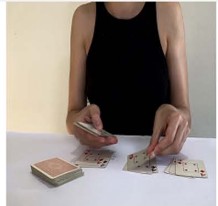 | **e) Distribute the cards on the table.**<br>NDUL - begins placing the cards on this side, with the shoulder in flexion and adduction while distributing the cards, gradually abducting the shoulder. The elbow performs flexion and extension, the forearm undergoes pronation and returns to a neutral position, the wrist flexes and extends with ulnar deviation, and a pinch grip is formed between the thumb and index finger, keeping the other fingers flexed. DUL - shoulder flexion and adduction, elbow flexion, forearm supination, wrist in a neutral position, and hand holding part of the cards with flexed fingers. | - Still using your D hand, hold part of the cards, and with your ND hand, pick up one card at a time to distribute it on the table, repeating the movements several times.<br>- Try to do it as quickly as possible.<br>- Pay attention to the movements your shoulders, elbows, wrists, and fingers are making, and then finish the task. |

**Fig 7. Description of the intervention as an example of a leisure activity - shuffle and deal the cards.** D, dominant; DUL, dominant upper limb; ND, non-dominant; NDUL, non-dominant upper limb.

established criterion for discontinuing the interventions assigned to a particular participant in the study will be solely at the request of the participant if they no longer wish to participate or if they do not follow the guidance to maintain their activities. To improve participant adherence to intervention protocols, participants will be contacted by telephone to confirm

participation in intervention sessions and, at the end of their participation in the study, each participant will receive a written feedback report of the results from the assessments conducted during the tests.

## Outcome

As the main outcome of the study, an improvement in the TEMPA score is expected, along with a decrease in the execution speed of tasks, indicating that participants experienced positive impacts on upper limb motor symptoms after interventions involving action observation, motor imagery, and brain-machine interface. As for secondary outcomes, improvements in motor symptoms are anticipated, evidenced by lower scores on the UPDRS-III and reduced time in the 9 Hole-Peg Test, as well as enhancements in cognitive symptoms indicated by increased scores on the PD-CRS. Additionally, it is expected that the interventions will also have a positive effect on occupational performance, leading to better performance and satisfaction in daily activities as assessed by the COPM. The averages of the assessments will be considered, with measurements for each outcome taken at T0 and T1, and at T2 regarding perceptions of changes in motor and cognitive conditions, as well as the COPM.

## Sample size

The sample size was estimated to detect a non-negligible and useful effect size in the primary outcome, TEMPA, corresponding to Cohen's d = 0.4 (f = 0.2), when applying ANOVA with a significant interaction effect (time*intervention) [30]. Considering two assessments, five groups, a significance level of 0.05 with 80% power, and accounting for an additional 19% margin for attrition, a total of 95 participants will be required, distributed into 19 participants per group.

Recruitment will be based on dissemination through electronic, radio, and print media, as well as contact with municipal health departments and associations as a strategy to achieve the target sample size.

## Statistical analysis

The results of qualitative variables will be expressed as absolute and relative frequency. Symmetric quantitative variables will be presented as mean and standard deviation, and asymmetric ones as median and interquartile range [P25 - P75]. The normality of the data will be checked using the Shapiro-Wilk test. Group characteristics will be compared using the Chi-square test, ANOVA, and/or false discovery rate (FDR) for the multiple comparisons between the 5 groups.

a)  A mixed repeated-measures ANOVA will be used, or, in the absence of normality and/ or in the presence of missing data, generalized estimating equations (GEE) models will be applied to assess the main effects of group and time, and the group*time interaction, with an unstructured correlation matrix. The goodness of fit will be assessed using the Quasi-Likelihood under Independence Model Criterion (QIC). The best-fitting distribution for the data (normal or gamma) will be evaluated using the AIC criterion, with identity or logarithmic link functions, respectively. Sidak's test will be used for multiple comparisons. The results will be presented as means and 95% confidence intervals (CI). Analyses will be conducted using SPSS software (IBM SPSS Statistics for Windows, Version 25.0. Armonk, NY: IBM Corp.). The significance level will be set at 0.05.

## Trial status

Participant recruitment began in February 2023. Data collection and data analysis are expected end by September 2026.

## Data management

The data collected throughout the study phases will be handled and stored in accordance with the General Data Protection Regulation (GDPR) 2018. The use of the study data will be controlled by the principal investigator. All data and documentations related to the study will be stored in accordance with applicable regulatory requirements and their access will be restricted to authorized personnel.

## Protocol amendments

Any modifications to the protocol that may impact the conduct of the study, including changes to the study objectives, study design, population, sample size, study procedures, or significant administrative aspects, will require a formal amendment to the protocol. The amendment will be developed in accordance with the protocol authors and submitted for attachment to the registration with the Research Ethics Committee of UFCSPA and the trial registry.

## Discussion

Recent randomized clinical trials (RCTs) involving AO and MI in PD aimed to evaluate the combination of MI with virtual reality techniques and physical exercise on motor function [31]; the combination of OA, MI and gait training in balance and freezing of gait [32]; whether AO with Sonification improves postural control [33]. Other studies have explored a home gait observation intervention to improve gait [34]; whether AO can improve freezing of gait and mobility in a group environment [3]; the combination of MI practice with physical walking practice in a single session [35]; whether MI has similar effects on mobility as relaxation rehabilitation [36], as well as improving mobility through motor physiotherapy [37] and risk of walking and falling [38].

Studies using AO and MI for UL interventions have explored: the reduction of bradykinesia in finger movements after a single AO session [39]; AO and combined AO + MI on hand range of motion [40]; AO + MI and home-based physical practice to enhance functional manual actions through timed dexterity and performance measures [41]; and the observation of movements while imitating action segments involving object-reaching and transferring [42]. These studies highlight the limited research on AO and MI interventions for UL motor rehabilitation, focusing on symptoms, manual functions, and daily activities while considering cognitive functions.

This study protocol is distinctive for its intensive daily sessions, contrasting with most protocols that use single or alternating sessions. Key features include first-person MI with auditory and tactile stimulation, third-person AO using healthy individuals' videos, and the incorporation of progressively challenging tasks. It addresses limitations of prior studies by combining AO and MI, focusing on upper limb outcomes, hand movements, and cognitive functions, which are underexplored [11,41].

The use of BMI technology in rehabilitation for neurological conditions has grown over the past decade, particularly in stroke cases [12,43,44]. BMI integrates robotic systems and brain signals to enhance rehabilitation [45]. In Parkinson's disease, this innovative approach, combined with motor imagery (MI) and action observation (AO), aims to improve upper limb cognitive and motor functions [10].

A differentiator of this protocol is the use of different MI approaches: one guided by the therapist with longer practice time, and another guided by the IMC software for a continuous rhythm and more repetitions in less time. Both methods instruct participants to visualize and feel the movement of the affected UL without physical movement, constructing

first-person images [46]. Functional MRI (fMRI) studies indicate that MI activates cortical [47] and subcortical motor areas, enhancing the learning of new manual skills [48]. Although people with PD present MI intensity similar to that of healthy individuals, they perform it more slowly [49]. BMI training may improve MI by engaging neural systems related to actual movement [50], although MI may not always be effective in PD, leading to defective neurofeedback [51].

Motor practice aimed at executing problematic daily tasks more productively, combined with challenging components (increasing task difficulty) and repetition of desired movement, can drive plasticity. The progression of this practice is key, starting with lower difficulty and progressing to higher as the movement is practiced productively [52] through repetition and task difficulty in terms of kinematics and kinetics [53,54]. It's well known that personalized task difficulty training yields superior learning outcomes compared to fixed difficulty increases [55].

This study protocol has some limitations. The intervention researcher will not be blinded due to the intervention's characteristics, especially the need to know the participant's motor and cognitive condition to direct the chosen activities. Therapists may also find it challenging to verify if mental training during MI is correct, as MI intervention depends on the individual's ability to imagine performing specific actions, except when using BMI.

In conclusion, the clinical trial protocol described here is easily replicable in practice by physical and occupational therapists for rehabilitating motor and cognitive symptoms in people with PD. MI and AO are low-cost and allow for personalized interventions based on the individual's needs in performing important activities, which can be done in settings other than the clinic. While BMI use requires professionals to acquire an exoskeleton, the protocol ensures the uniqueness of the interventions and stands out as the first, to our knowledge, to involve people with PD. The expected results and findings from this study is only representative for this sample, and future trials with larger sample sizes and at other geographical locations will be necessary to assess the effectiveness of the interventions.

## Supporting information

**S1 Checklist. SPIRIT checklist.**
(PDF)

**S1 Graphycal Abstract. Trial timeline.**
(TIF)

**S1 Video. Example of personal care activity - organize the week's medication.**
(MP4)

**S2 Video. Example of productivity activity - hang the clothes on the clothesline.**
(MP4)

**S3 Video. Example of a leisure activity - shuffle and deal the cards.**
(MP4)

**S1 File. Prior approval from the ethics body (original).**
(PDF)

**S2 File. Prior approval from the ethics body (translated).**
(PDF)

**S3 File. Protocol registration ClinicalTrials.**
(PDF)

**S4 File. Study protocol translated.**
(PDF)

**S5 File. Original study protocol.**
(PDF)

## Author contributions

**Conceptualization:** Kátine Marchezan Estivalet, Natália Lopes Mazzilli, Luis Fernando Ferreira, Fernanda Cechetti.

**Methodology:** Kátine Marchezan Estivalet, Tatiana Salayaran de Aguiar Pettenuzzo, Natália Lopes Mazzilli, Fernanda Cechetti.

**Project administration:** Fernanda Cechetti.

**Supervision:** Luis Fernando Ferreira, Fernanda Cechetti.

**Validation:** Fernanda Cechetti.

**Visualization:** Kátine Marchezan Estivalet, Tatiana Salayaran de Aguiar Pettenuzzo, Luis Fernando Ferreira.

**Writing – original draft:** Kátine Marchezan Estivalet, Tatiana Salayaran de Aguiar Pettenuzzo, Natália Lopes Mazzilli.

**Writing – review & editing:** Kátine Marchezan Estivalet, Luis Fernando Ferreira, Fernanda Cechetti.

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
