## [Decision Letter · Decision Letter 0]

5 Jan 2025

PONE-D-24-50579The use of brain-machine interface, motor imagery, and action observation in the rehabilitation of individuals with Parkinson's disease: a protocol study for a randomized clinical trialPLOS ONE

Dear Dr. Ferreira,

Thank you for submitting your manuscript to PLOS ONE. After careful consideration, we feel that it has merit but does not fully meet PLOS ONE’s publication criteria as it currently stands. Therefore, we invite you to submit a revised version of the manuscript that addresses the points raised during the review process.

We look forward to receiving your revised manuscript.

Kind regards,

Imre Cikajlo, Ph.D.

Academic Editor

PLOS ONE

Reviewers' comments:

Reviewer's Responses to Questions

**Comments to the Author**

1. Does the manuscript provide a valid rationale for the proposed study, with clearly identified and justified research questions?

Reviewer #1: Yes

Reviewer #2: Yes

2. Is the protocol technically sound and planned in a manner that will lead to a meaningful outcome and allow testing the stated hypotheses?

Reviewer #1: Yes

Reviewer #2: Yes

3. Is the methodology feasible and described in sufficient detail to allow the work to be replicable?

Reviewer #1: Yes

Reviewer #2: Yes

4. Have the authors described where all data underlying the findings will be made available when the study is complete?

Reviewer #1: Yes

Reviewer #2: Yes

5. Is the manuscript presented in an intelligible fashion and written in standard English?

Reviewer #1: Yes

Reviewer #2: Yes

6. Review Comments to the Author

You may also provide optional suggestions and comments to authors that they might find helpful in planning their study.

Reviewer #1: This manuscript presents a study protocol to conduct a single-blinded, randomized clinical trial for comparing 5 experimental groups in rehabilitation with Parkinson's disease. The study was registered in clinicaltrials.gov with a valid NCT number, and approved by the respective Ethics/IRB board. While the objectives and timeliness of this project appear sound and convincing, some comments appear below:

(a) Statistical Analysis: For the multiple comparisons, it might be worthwhile to use the false discovery rate (FDR) adjustments over the Tukey's test, since there are a lot of comparisons to be made between the 5 groups.

(b) Statistical Analysis: A GEE will be conducted to handle data generated from the longitudinal design (multiple time-points). Any reason, why the usual linear mixed model won't be used? What about assessments of the goodness of fit from the GEE fits, or the linear mixed model fits.

(c) Conclusions/Discussion: This section should mention that the expected results and findings from this study is only representative for this sample of patients, and future trials with larger sample sizes and at other geographical locations will be necessary to assess the effectiveness of the interventions.

Reviewer #2: The article is really well written, and it doesn’t almost need formal changes. I’ve only a very few formal criticisms: Authors underlined too frequently the fact that their protocol is very detailed, and it is the first one to be presented in a so detailed manner. One time is enough, I think.

Table 1: Open medication bottle and the following tasks with medications. Why instructions for AO and MI are different as far as the use of dominant and not dominant hand is concerned? This would make AO and MI not comparable. Moreover, the description seems to me contradictory to the statement at page 26, lines 407-8.

Page 27, lines 448-450 “Recruitment based on media outreach and contact with municipal health departments and associations as an enrollment strategy to reach the target sample size.” It seems to me that some words are lacking in this sentence.

On the other hand, I’d like to express a few ideas of mine about the protocol content. The Authors are not obliged to change the protocol according to my ideas, but I’d like to give them some suggestions to think of, and maybe better motivate their choices in the text.

Selection of participants: “Additionally, those assigned to the BMI groups must not be using deep brain stimulation (DBS) devices.” This should be true for all groups. The presence of a DBS device or not represents a great difference between participants with PD for a lot of reasons.

Evaluation of participants: ““How would you rate the severity of your cognitive functions (memory, attention, concentration, organization”. Are the Authors sure that the patients will make a difference between attention and concentration? What’s the difference between attention and concentration according to the Authors?

Experimental treatment, motor imagery: “The activities will be narrated by the therapist during the intervention, guiding the participant - eyes closed and seated - to imagine executing the action from a first-person perspective, without the presence of motor symptoms”. How are the Authors sure that the participants affected by PD will be able to imagine the movement without symptoms? Maybe they should at least ask the participant whether they succeeded or not.

It's really a great limitation the fact that the follow-up evaluation will be performed only by phone and only using the COPM. Why did the Authors choose not to re-examining the enrolled participants in flesh?

7. PLOS authors have the option to publish the peer review history of their article (what does this mean? ). If published, this will include your full peer review and any attached files.

**Do you want your identity to be public for this peer review?** For information about this choice, including consent withdrawal, please see our Privacy Policy .

Reviewer #1: No

Reviewer #2: **Yes: ** Elisabetta Farina

---

## [Author Response · Author response to Decision Letter 1]

13 Jan 2025

Response to Reviewers

Dear Academic Editor and Reviewers,

In response to your guidance, I address each point raised for corrections to the article. I would like to thank you in advance for your valuable contributions to improving the protocol.

For Reviewer #1:

a) We have revised the statistical method as suggested, where the False Discovery Rate (FDR) will be used for multiple comparisons among the five groups (lines 462).

b) The choice of GEE is to avoid losing potential cases in the event of follow-up loss. The goodness of fit will be assessed using the Quasi-Likelihood under Independence Model Criterion (QIC) (lines 467-468).

c) We agree with your comments and have included the suggested information as the last sentence of the article's conclusion (page 32, lines 568-571): “The expected results and findings from this study are representative only of this sample, and future trials with larger sample sizes and conducted in other geographical locations will be necessary to assess the effectiveness of the interventions.”

For Reviewer #2:

• I have edited the text to avoid repetition regarding the description of the protocol by removing the word "detailed" in the objective section of the abstract (line 34), in the last paragraph of the introduction (line 105), and in the first paragraph of the study design section in the methods (line 112).

• Table 1 (pages 17-19):

The verbal command directed at the participant during MI is correct according to the images. However, there was an error in the technical description of the movements performed during AO, which has now been corrected in the article. We appreciate the observation. It is worth noting that the technical description is not provided to the participant, as they only observe the videos without any additional stimuli. This description is exclusively for therapists when creating the videos, ensuring proper consideration of the movements for each activity stage. Although the person in the video is right-handed, bimanual activities can be performed with either the dominant or non-dominant hand, while unilateral activities are carried out using the side where motor symptoms are present. Therapists adjust the MI guidance accordingly, instructing participants to use either their right or left hand depending on the side affected by symptoms, using accessible language.

• Page 27, lines 448-450:

I have revised the sentence to make it complete, including the types of media used for research dissemination (lines 453-455).

• Selection of participants:

I have corrected the selection criteria (page 7, lines 157-158), specifying that DBS was considered an exclusion criterion for all groups, precisely for the reason mentioned.

• Evaluation of participants:

The intention is to illustrate cognitive functions (memory, attention, concentration, organization), as participants may not be familiar with the concept of cognition or cognitive functions. However, since the assessment does not focus on any specific function but rather evaluates a general state, we do not believe this presents an issue. Nonetheless, we can incorporate real-life examples, such as:

o “If you are watching television and someone asks you for a favor, can you focus only on the television or the request? Can you remember the favor afterward, or do you forget it quickly?”

o “Is it easier for you to remember what you need to buy at the market or memorize someone’s phone number?”

• Experimental treatment, motor imagery:

Since we will use the Kinesthetic and Visual Imagery Questionnaire (KVIQ-10) during screening, we can already evaluate the participant's imagery ability, including image clarity and sensation intensity. However, we agree it is essential to ask participants afterward whether they were able to imagine themselves, and we have added this information to the protocol (page 12, lines 274-277).

• We removed the section where we mentioned conducting the COPM by phone as a limitation of the study (lines560-563). The COPM is a semi-structured interview in which individuals assess their own perceptions of changes in occupational performance. This format allows for remote application, making it more convenient for participants by eliminating the need for an additional in-person meeting for data collection.

I would like to highlight that I have corrected the formatting of the figures (Fig 1 and Fig 2), placing the captions below the titles (pages 7-8).

---

## [Decision Letter · Decision Letter 1]

21 Feb 2025

The use of brain-machine interface, motor imagery, and action observation in the rehabilitation of individuals with Parkinson's disease: a protocol study for a randomized clinical trial

PONE-D-24-50579R1

Dear Dr. Ferreira,

We’re pleased to inform you that your manuscript has been judged scientifically suitable for publication and will be formally accepted for publication once it meets all outstanding technical requirements.

Kind regards,

Emiliano Cè, Ph.D.

Academic Editor

PLOS ONE

Additional Editor Comments (optional):

Reviewers' comments:

Reviewer's Responses to Questions

**Comments to the Author**

1. Does the manuscript provide a valid rationale for the proposed study, with clearly identified and justified research questions?

Reviewer #1: Yes

2. Is the protocol technically sound and planned in a manner that will lead to a meaningful outcome and allow testing the stated hypotheses?

Reviewer #1: Yes

3. Is the methodology feasible and described in sufficient detail to allow the work to be replicable?

Reviewer #1: Yes

4. Have the authors described where all data underlying the findings will be made available when the study is complete?

Reviewer #1: Yes

5. Is the manuscript presented in an intelligible fashion and written in standard English?

Reviewer #1: Yes

6. Review Comments to the Author

You may also provide optional suggestions and comments to authors that they might find helpful in planning their study.

Reviewer #1: The authors were able to address my previous round of comments with great satisfation. I have no further comments in this round.

7. PLOS authors have the option to publish the peer review history of their article (what does this mean? ). If published, this will include your full peer review and any attached files.

**Do you want your identity to be public for this peer review?** For information about this choice, including consent withdrawal, please see our Privacy Policy .

Reviewer #1: No

---

## [Editor Report · Acceptance letter]

PONE-D-24-50579R1

PLOS ONE

Dear Dr. Ferreira,

I'm pleased to inform you that your manuscript has been deemed suitable for publication in PLOS ONE. Congratulations! Your manuscript is now being handed over to our production team.

Kind regards,

on behalf of

Prof. Emiliano Cè

Academic Editor

PLOS ONE